# Changing the Level of Education and Career Choice Depending on the Socioeconomic Status of the Family: Evidence from Azerbaijan

Natavan Namazova [1,2]

1  International Trade, Logistic and Marketing Department, Azerbaijan Technical University,
   Baku AZ1001, Azerbaijan; natavan.namazova@aztu.edu.az or natavan.namazova@unec.edu.az
2  Centre for Studies on European Economy (AIM), Azerbaijan State University of Economics (UNEC),
   Baku AZ1001, Azerbaijan

**Abstract:** Education plays an important role in the fight against poverty and contributes to the formation of human capital by increasing the knowledge and skills of the individual. It increases the educational opportunities for future generations, provides more opportunities to participate in public life, and makes a significant contribution to social development. Education is influenced by various factors. One of the main factors influencing education is the socio-economic conditions of family life. This study explores the relationship between the socio-economic status of the family and the level of education in Azerbaijan. To this end, the influence of two main independent variables, namely, the influence of family elders and family income, on the level of education of an individual was studied. As a result of the study, it was found that mothers compared to grandparents and grandparents compared to fathers have a positive impact on the level of education of children. It was also found that household income is a determining factor in choosing a risky career and that children from families with an upper middle income are especially more positive than those from low income families.

**Keywords:** education level; household income; family elders; risky career; income

## 1. Introduction

It is generally accepted that education, which breaks the cycle of ignorance and poverty, is directly related to the level of development of countries today [1]. For this reason, various studies of education in many branches of science are found in literature.

Unfortunately, individuals born with the same characteristics differ from each other, as they are affected by inequalities in the structure of the society around them and the socioeconomic structure of their families [2–4]. Specifically, a child born into a poor family always struggles with the difficulties of poverty. This is where education comes into play and opens new doors for people. Education is one of the important elements in fighting poverty and improving income distribution, especially due to its ability to narrow the income gap [5–7].

Education has an important role in the fight against poverty and contributes to the formation of human capital by increasing the knowledge and skills of the individual [7,8]. Education expands an individual's efforts to find a job, increase productivity in the workplace, and earn higher incomes. All levels of education also increase productivity [9]. Since the individual who succeeds in raising his/her income level will have a good job, his/her social status will also increase [7,10–13].

Another reason for the importance given to education in the fight against poverty is that children from poor and low income families are deprived of educational opportunities, and this leads to poverty becoming persistent from generation to generation [2,7,14–16].

Today, when we compare developed and industrialized societies with developing countries, we see that the most important difference between them is the level of education and knowledge. Education increases the productivity of the workforce, improves income distribution, strengthens physical and mental health, provides longer life, increases the opportunity for future generations to receive education, provides more opportunities for participation in social life, and makes a significant contribution to social development [17,18]. The individual who feels productive and happy thanks to education will make a higher contribution to the development and growth of the country's economy.

Thus, education is an important phenomenon, both in terms of ensuring the continuity of the social system and in terms of contributing to the development of the individual and society through the production of new knowledge, as well as of acquired skills and values. Therefore, the main goal of this study is to investigate the predicted relationship between the socioeconomic structure of the family and education in Azerbaijan.

## 2. Socioeconomic Impact of Education

Education allows us to increase human capital. Human capital refers to the knowledge and skills possessed by the workforce, which enables personal and social development and increases economic welfare. Increasing human capital accelerates social development by improving the skills and productivity capacity of human resources [19–22]. For example, a tailor with a higher education level can attain a higher level of production efficiency than a tailor with a lower education level.

If we consider the relationship between education and the economy, we can see that this relationship takes place in different dimensions. Various aspects of this effect could be ranked as production effect, income effect, expenditure effect, and development effect.

It is possible to make different prioritizations within these effects. That is, according to economists, while financial factors are determinant of the demand for education, according to sociologists, the status of the individual's family and the social environment surrounding the individual are also important factors.

Production effect. As we know, labor is one of the production factors and it is of great importance in the economy. Therefore, today, a qualified workforce is a phenomenon that all societies want to have, because a trained workforce is important for all aspects of the development and progress of countries. There are various factors in the market that condition the competition that countries want to gain priority. One of these factors is cheap prices. Providing cheap production costs is more related to productive labor and quality. Education improves the working habits of individuals and increases their productivity in the production process. Increasing the development of human resources helps the country to form and expand the necessary environment for domestic and foreign investments [8,23–25].

It should be noted here that the skills acquired as a result of basic education may be sufficient to produce products according to simple technology and ensure minimum quality. However, to increase the productive effect of education, individuals must acquire high skills, be ready for entrepreneurship, and feel free to choose a risky career. At this time, the output of individuals into society will increase. For this reason, today, skilled labor is expressed as human capital.

Income effect. The income effect of education can be explained by the fact that the educated person finds a job with a higher salary. In other words, one of the factors affecting the relationship between labor and income is the level of education [25–28], because educated people contribute to the efficiency of production and increase the added value in the economy. Human resources involved in the production process are divided into two groups, qualified and unqualified. Neither group can be substituted for the other, except for routine work. Therefore, the wages of qualified personnel are higher than those of other employees [7,29,30].

Increasing the level of education of people leads to their receiving more income during the working period, as well as a higher income after retirement. In other words, the failures

people experience in education throughout their lives may cause people to work for lower wages, find jobs in worse conditions, and continue to live on low incomes after retirement.

On the other hand, the increase in the level of education will not only increase the income of the people at that time, but also affect the shaping of future generations. Consequently, positive change in the field of education today will lead to better education and higher income levels for future generations [31,32].

Cost effect. A strong and effective education system requires serious investments in the economy. The expenditures to be made for the education of the individual are directly related to personal, cultural, institutional, sociodemographic, and economic variables [32–37]. These expenditures can be made by the government, the private sector, and families. For example, certain annual payments (teacher salaries, expenditure on consumer goods for use, etc.) by the government and private sector can be included here. Families also have a large share in the expenditures allocated to the education of individuals. Family income directly and indirectly affects the education of the child [38]. Direct effects can include books, uniform, and handling fees. Indirect effects include the possibility of residing in developed regions and attending good schools or self-development courses [39]. Sometimes, families' expenditures for the education of their children can create a great economic burden for the families [40]. In other words, the socioeconomic status of the family can sometimes be the most determining factor in the demand for education, due to this burden.

Development effect. The education level of people affects the level of savings, investment rates and investment areas, employment status, income distribution, and many macro variables in the economy [40–44]. At the same time, the successes achieved in educational life increase the productivity of individuals by raising their living standards, increase the opportunities for growth and innovation, prevent social corruption and crime, and decrease public health expenditures and social assistance needs [45–48].

### 3. Effect of Family Factor on Education

The importance of increasing the educational opportunities of the poor as a strategy for reducing poverty and ensuring education's sustainability by affecting different areas of society is emphasized in World Bank research, because poverty and low income levels affect families' opportunities to secure the financial resources they need for their children's education. This situation negatively affects the investments of low income families in education and causes the transfer of income inequalities between generations [49,50].

Before examining the relationship between the family factor and education, let us take a look at the different factors that determine the education demand of individuals. In general, the following factors are important factors affecting the education level of individuals [31,38,51]:

- Social environment;
- Family income;
- Government spending on education;
- Academic ability;
- Preferences or personal tastes;
- The possibility of finding a job;
- Future income.

Each of these elements affects the desire and ability of an individual to obtain an education and influences the level of education in society. However, within the framework of this article, only the influence of the social environment or family income created by the family factor, which has an important influence among these factors, was investigated, because the family factor determines whether a child will receive an education and how long he/she will continue his/her education, influences his/her success, and plays a decisive role in the use of educational opportunities.

A review of the literature shows that this relationship has been discussed by many researchers. In particular, it is emphasized that the level of education of parents and family

income are important factors in the participation of the child in education. Education is highly correlated with many socioeconomic variables [36,52].

The definition of the relationship between education and family income needs to be clarified for several specific reasons. Since education affects personal incomes, the level of poverty and income distribution in the future can be regulated by today's educational opportunities. For this reason, let us first examine the relationship between education and family income, which has been widely explored in the literature. One of these studies was carried out by Christofides, Cirello, and Hoy [53]. The researchers examined educational participation status by income group in Canada from 1975 to 1993. In this study, they analyzed the higher education attendance status of children from high and low income families. According to this study, in 1975, the rate of attendance of children from the richest families in higher education was more than three times higher than that of children from the poorest families, while in 1993 this rate was 1.6 times higher.

Another study was carried out by Becker (1993) in the USA. Becker touched upon the impact of the family on human capital and stated that the future income of the children of families whose average income is 20% higher than their peers in the USA is approximately 6% higher than that of their peers. Wealthy families allocate more resources to their children's education, while children from poor families may not be as responsive to this type of investment as their children tend to live away from home for various reasons (physical, psychological, etc.) [54].

A study was conducted on the relationship between the income level of families and children's access to high schools by Çelikkol and Avcı (2017) [28,55]. In this study, it was found that, as the income level of the families increased, the level of success of the children in the central exams they sat during the transition to secondary education also increased.

The type and level of association between family income and child participation in education also differ depending on the type of funding for education and the level of education in the countries. Acemoglu and Pishke (2000) stated in their study in the US that family income is a strong factor in a child's participation in further education. According to the results of the research, a 10 percent increase in family income can increase the probability of attending a four-year university course by 1.4 percent [56].

In the study conducted by Şemin on children who are mentally talented but unsuccessful at school, it was concluded that the sociocultural level of the family is relevant in school success [57]. Accordingly, the families with the highest level of failure (73%) are those who are not well off; these families are low income workers and small shopkeepers. The higher the socioeconomic level, the lower the failure rate: the failure rate is 24% among children from middle-class families and 3% among those from well-off families.

Maitra (2003) tried to explain the effects of personal and household characteristics on education demand in Bangladesh. The researcher states that increased family income results in increased participation in education [58].

Corak, Lipps, and Zhao (2003) found in their study that children from high income families are more likely to attend university [59]. Blanden, Gregg, and Machin (2003) state that there are inequalities in participation in education according to income groups in their study [60].

As has been observed in prior studies, there are many different factors that determine the level and quality of education that individuals will receive [61]. Among these, it is seen that one of the most influential factors is parents. Fathers have a great role in the social–emotional development of children, assuming their responsibilities and roles in society and in the family [62–64]. The father has a great role in the social–emotional development of the child [65]. Handa et al. (2004) investigated the effects of variables such as the education level of the parents, wealth, gender, rural area, number of individuals in the household, and having siblings on the education level of the individual. In particular, they concluded that the education level of the father has an effect on the education of the children [66].

In another study conducted in 2014, the effect of families' socioeconomic characteristics on student achievement was examined, and as a result of the research, it was found that the



socioeconomic situation of the families was highly influential in the success of the children in the exams they entered [67].

Dumas and Lambert (2005) investigated the relationship between education and family background characteristics. They used a survey conducted in Senegal in 2003. Contrary to the expected results, it was concluded that the education level of the father was more effective than the education level of the mother in influencing the education level of the individuals [68].

Every child who grows up with his/her mother and receives education under her influence achieves appropriate physical, psychological, and social development. These children generally enjoy childhood and can operate well within their environment [69].

The roles of parents in the education of the child support each other. Therefore, it is necessary for both of them to participate in the education process [70].

The socioeconomic status of families has an impact not only on the educational status of individuals, but also on their career choices. Specifically, children from families with high socioeconomic status tend to choose careers with higher risks and higher incomes [71]. The children of families with low socioeconomic status may tend to prefer a career with no risk, job security, and a lower income. For example, Raftery and Hout (1993) examined the repercussions of the egalitarian reforms implemented in the Irish upper secondary education system in 1967 [72]. As a result of the reforms designed to reduce inequality, the participation of children from low-income families in vocational high schools or evening schools where they could work part-time increased. In other words, poor families had previously made a rational choice and had not continued to invest in their children's education. Children from low income groups who do not continue with higher education have acquired professions that are not valued by society and are subject to low wages [73,74]

Another research on risky career choices was conducted by Saks and Shore (2005) [75]. In their study based on US data, they found that wealthier individuals were more likely than others to choose riskier branches such as business. Caner and Ökten (2010) included data obtained from approximately 40,000 students in 2002 in their research in Turkey. This study revealed that family income, fathers' self-employment, and social security status had strong effects on students' risky career choices [76].

## 4. Hypotheses

The examples given above show that household income and family elders can affect the education level of children. For this reason, the effect of the socioeconomic status of families on the education level of individuals in Azerbaijan was investigated in this study. For this purpose, the effects of two main independent variables, namely, the influence of family elders and household income on the education level of the individual, were examined.

The main hypotheses of the study are as follows:

**Hypothesis 1.** *The educational level of individuals varies according to the socioeconomic conditions of the family.*

**Hypothesis 1a.** *Individual education level varies depending on the influence of family members.*

**Hypothesis 1b.** *Individual education level varies depending on household income.*

**Hypothesis 2.** *An individual's risky career choice depends on the socioeconomic status of the family.*

**Hypothesis 2a.** *An individual's risky career choice depends on the influence of family members.*

**Hypothesis 2b.** *An individual's risky career choice depends on household income.*

## 5. Material and Methods

The study used primary data collection techniques as a scientific research method. The research survey was sent to 920 respondents in Baku. Considering that in Azerbaijan children generally grow up in the same house as their grandparents, an evaluation of the impact of grandparents on individuals was also attempted. For this reason, only the answers of those who grew up with their grandparents and whose elders had a high education level (at least a bachelor's degree) were included in the study. Accordingly, 504 responses out of 920 were accepted for analysis. The author conducted the survey between January and December 2022.

### 5.1. Findings Regarding the General Characteristics of the Survey Participants

In this section, the data obtained regarding the general characteristics of the survey participants, such as household income, choice of risky profession, influence of family members and educational level of individuals, are presented.

According to the results presented in Table 1, 33.9% of the participants have a high school diploma, 27.2% have a bachelor's degree, 22.0% have a master's degree, and 16.9% have a doctoral degree. These results indicate that the majority of the participants have a high level of education.

**Table 1.** Sample distribution and percentages.

| | Number | Percent (%) | | Number | Percent (%) |
|---|---|---|---|---|---|
| **Participants' education level** | | | **Influence of family elders** | | |
| High school | 171 | 33.9 | Father | 167 | 33.1 |
| Bachelor's | 137 | 27.2 | Mother | 201 | 39.9 |
| Master's | 111 | 22.0 | Grandparents | 136 | 27.0 |
| Ph.D. | 85 | 16.9 | | | |
| **Risky profession choice** | | | **Household income** | | |
| No | 241 | 47.8 | Low income | 139 | 27.6 |
| Yes | 155 | 30.8 | Lower middle income | 174 | 34.5 |
| Other | 108 | 21.4 | Upper middle income | 115 | 22.8 |
| | | | High income | 76 | 15.1 |
| **N = 504** | | | | | |

Other data in the study are related to the proportion of family elders with a high level of influence. In other words, according to the given data, the high influence of fathers on the participant is approximately 33.1%. The rate of mothers with high influence is 39.9%, while the rate for grandparents is 27%.

Of the respondents, 47.8% said that the reason for choosing a profession was that they would easily find a job, would always generate an income, and would not be afraid of being fired. Moreover, 30.8% of the participants stated that they considered their profession risky but still chose it, and 21.4% of the participants said they chose it for other reasons.

Another difference noted among the respondents related to the income level of the households. While the majority of the participants (34.5%) saw themselves in the lower middle income group, 27.6% of the participants classified themselves as being in the low income group. Only 22.8% and 15.1% of the respondents represented themselves as having an upper middle income and high income, respectively.

### 5.2. Data Analysis

The variables were subjected to one-way analysis of variance (ANOVA) tests. ANOVA analysis was performed using IBM SPSS Statistical Version 26. Before the ANOVA analysis, it was determined whether the data showed a normal distribution. For this purpose,

skewness and kurtosis values were found first. The data presented in Table 2 show that the skewness and kurtosis values of all variables ranged from −1 to +1. Based on these values, we were able to proceed with the ANOVA analysis.

**Table 2.** Skewness and kurtosis coefficients of the variables.

| Statistics | | Influence of family members | Household income |
|---|---|---|---|
| N | Valid | 504 | 504 |
| | Missing | 0 | 0 |
| Skewness | | 0.538 | 0.426 |
| Std. error of skewness | | 0.331 | 0.281 |
| Kurtosis | | −0.764 | −0.390 |
| Std. error of kurtosis | | 0.503 | 0.128 |

Then, Cronbach's alpha analysis was used to measure the reliability of the scales used in the research. As a result of the analysis, the Cronbach's alpha values were found to be 0.793 and 0.775, that is, a value higher than 0.70, which indicates that the inter-item agreement is high.

## 6. Results and Findings

### 6.1. Verification of Hypothesis 1

In order to prove Hypothesis 1a, the education level of individuals was analyzed from both dimensions, namely, the high influence of family elders and risky profession choice was assessed.

Table 3 reveals the ANOVA results related to Hypothesis 1a. When the ANOVA table is examined, it is seen that the "Sig." value is less than 0.05. Based on this result obtained by performing a one-way analysis of variance, Hypothesis 1a is accepted.

**Table 3.** Results of Analysis of Variance in relation to the level of individual education and influence of family members.

| ANOVA | | | | | |
|---|---|---|---|---|---|
| Influence of family elders | | | | | |
| | Sum of Squares | Df | Mean Square | F | Sig. |
| Between groups | 28.147 | 2 | 15.419 | 26.204 | 0.000 |
| Within groups | 462.909 | 502 | 0.705 | | |
| Total | 491.056 | 504 | | | |

In other words, a significant relationship was found between the level of individual education and the high influence of family elders.

A Tukey's post-hoc test was used to determine the direction of the relationship between these variables (Table 4). According to the Tukey's post hoc test results, the main difference is first seen between father and grandparents. Namely, the grandparents had a positive effect on the individual education level compared to the fathers. The other difference was observed between mothers and grandparents. Mothers had a more positive effect on the education level of individuals compared to grandparents.

**Table 4.** Tukey HSD regarding the level of individual education and influence of family members.

| **Multiple Comparisons** | | | | | | |
|---|---|---|---|---|---|---|
| Dependent Variable: Level of individual education | | | | | | |
| Tukey HSD | | | | | | |
| | (J) household income | Mean Difference (I-J) | Std. Error | Sig. | 95% Confidence Interval | |
| | | | | | Lower Bound | Upper Bound |
| Father (I) household income | Mother | −0.78034 | 0.95724 | 0.346 | −0.1864 | −0.4472 |
| | Grandparents | −0.90134 * | 0.54692 | 0.000 | −0.2734 | −0.5240 |
| Mother | Father | 0.78034 | 0.95724 | 0.346 | 0.4472 | 0.1864 |
| | Grandparents | 0.53361 * | 0.13475 | 0.001 | 0.1477 | 0.7162 |
| Grandparents | Father | 0.90134 * | 0.54692 | 0.000 | 0.5240 | 0.2734 |
| | Mother | −0.53361 * | 0.13475 | 0.001 | −0.7162 | −0.1477 |

* The mean difference is significant at the 0.05 level.

Verification of Hypothesis 1b.

To confirm H1b, ANOVA analysis was performed in terms of individuals' education level and household income. The results obtained are presented in Table 5.

**Table 5.** ANOVA results regarding the level of individual education and household income.

| **ANOVA** | | | | | |
|---|---|---|---|---|---|
| Household income | | | | | |
| | Sum of Squares | Df | Mean Square | F | Sig. |
| Between groups | 83.619 | 3 | 96.533 | 53.220 | 0.039 |
| Within groups | 472.640 | 501 | 0.753 | | |
| Total | 556.259 | 504 | | | |

According to the result obtained by ANOVA analysis, the significance value is higher than 0.05 and H1b was not accepted.

### 6.2. Verification of Hypothesis 2

The relationship between the choice of risky profession and the socioeconomic status of the family is shown in Tables 6 and 7.

**Table 6.** ANOVA results regarding choice of risky profession and influence of family members.

| **ANOVA** | | | | | |
|---|---|---|---|---|---|
| Choice of risky profession | | | | | |
| | Sum of Squares | df | Mean Square | F | Sig. |
| Between groups | 33.792 | 2 | 24.063 | 28.767 | 0.094 |
| Within groups | 291.146 | 502 | 0.623 | | |
| Total | 324.938 | 504 | | | |

**Table 7.** ANOVA results regarding choice of risky profession and household income.

| **ANOVA** | | | | | |
|---|---|---|---|---|---|
| **Choice of risky profession** | | | | | |
| | Sum of Squares | Df | Mean Square | F | Sig. |
| Between groups | 55.482 | 3 | 4.170 | 45.336 | 0.000 |
| Within groups | 459.148 | 501 | 0.725 | | |
| Total | 514.630 | 504 | | | |

Verification of Hypothesis 2a.

As can be seen from Table 6, the "sig." value of ANOVA is higher than 0.05. This means that any group that differs in their choice of risky occupation is not significantly different from the overall group mean.

Therefore, within the scope of these results, H2a was rejected.

Verification of Hypothesis 2b.

The final ANOVA analysis revealed the relationship between the choice of risky profession and household income.

As can be seen from Table 7, the "Sig." value is equal to 0.000, which is less than 0.05. This indicates that the choice of risky profession by an individual differs depending on the household income. Thus, H2b was accepted based on the results obtained.

The Tukey's post hoc test revealed that a relationship emerged between two variables (Table 8). These variables are the low income group and the upper middle income group. In other words, individuals in the upper middle income group are more inclined to choose risky occupations than individuals from the lower income group.

**Table 8.** Tukey HSD results regarding choice of risky profession and household income.

| **Multiple Comparisons** | | | | | | |
|---|---|---|---|---|---|---|
| Dependent Variable: Choice of risky profession | | | | | | |
| Tukey HSD | | | | | | |
| (I) household income | (J) household income | Mean Difference (I-J) | Std. Error | Sig. | 95% Confidence Interval | |
| | | | | | Lower Bound | Upper Bound |
| Low income | Lower middle income | −0.51502 | 0.19374 | 0.138 | −0.8619 | −0.4073 |
| | Upper middle income | −0.69076 * | 0.15091 | 0.001 | −0.1143 | −0.5241 |
| | High income | −1.57800 | 0.01032 | 0.880 | −1.4262 | −0.8261 |
| Lower middle income | Low income | 0.51502 | 0.19374 | 0.138 | 0.4073 | 0.8619 |
| | Upper middle income | 0.45532 | 0.15735 | 0.120 | 0.3047 | 0.7081 |
| | High income | −0.63691 | 0.12292 | 0.092 | −0.7114 | −0.1015 |
| Upper middle income | Low income | 0.69076 * | 0.15091 | 0.001 | 0.5241 | 0.1143 |
| | Lower middle income | −0.45532 | 0.15735 | 0.120 | −0.7081 | −0.3047 |
| | High income | −0.65231 | 0.11462 | 0.072 | −0.2256 | −0.7684 |
| High income | Low income | 1.57800 | 0.01032 | 0.880 | 0.8261 | 1.4262 |
| | Lower middle income | 0.63691 | 0.12292 | 0.092 | 0.1015 | 0.7114 |
| | Upper middle income | 0.65231 | 0.11462 | 0.072 | 0.7684 | 0.2256 |

* The mean difference is significant at the 0.05 level.

## 7. Discussion and Conclusions

Today, one of the issues to which society attaches great importance is ensuring the sustainability of society and reducing the risk of poverty that individuals may face. To do this, education must be achieved in society and appropriate conditions must be created for the self-development of the individual. To improve the level of education of every person in Azerbaijan, various services are offered on the market by both the state and the private sector. However, in order for individuals to receive these services or to improve themselves, the impact of the environment surrounding them, especially the socioeconomic status of the family, should be taken into account.

Our research revealed important factors. Specifically, it found that, although the level of education of individuals participating in the survey is not influenced by household income, it is strongly influenced by family elders. The reason why the level of education of people does not depend on family income can be explained by the opportunities for free education offered by the state. Specifically, basic education in Azerbaijan is compulsory and is provided by the state to all children free of charge. Moreover, university tuition fees are covered by the state for individuals with high exam scores.

Research results show that success at the educational level of an individual depends on social rather than economic conditions. This effect is especially noticeable in mothers. In fact, no relationship was found between the influence of mothers and fathers, but the influence of mothers on the education of children turned out to be more positive than the influence of grandparents. On the other hand, it was found that grandparents more effectively increase the level of education of children compared to fathers. This result shows that, in order to further improve the level of education of children in the country, the development of those who influence the education of children, especially mothers and grandparents, should always be supported. Self-developed family elders will encourage children to receive a better education and help them realize themselves in the future and gain new knowledge and skills.

Another result obtained in the study is related to the choice of individuals to enter risky professions. According to the results, a difference was observed only between people who grew up in low income families and children from upper middle income families. While children from low income families were not very positive about risky professions, children from upper middle income families showed the opposite approach. When choosing a profession, children from low income families strive to receive a stable income and are not afraid of being fired. However, children from families with upper middle and high income levels are far from sharing these concerns and tend to work in professions that are more uncertain and do not promise stability. As we know, in order for the economy to develop as a free market economy, people must be more entrepreneurial and ready to take risks. In order for individuals with an entrepreneurial spirit to grow up in Azerbaijan, special programs should be prepared for individuals, especially children from low income families, and efforts should be made to instill in them the notion that risky professions should not be avoided when choosing a profession. This will help Azerbaijan to develop high-quality human resources, to produce highly educated people suitable for production needs, to increase the country's income and, as a result, to increase its competitiveness in world markets.

**Funding:** This research received no external funding.

**Institutional Review Board Statement:** The study was conducted in accordance with the Declaration of Helsinki and approved by the Institutional Review Board (or Ethics Committee) of Economic Think (protocol code 301504 and date of approval 11 January 2022).

**Informed Consent Statement:** Informed consent was obtained from all subjects involved in the study.

**Data Availability Statement:** Not applicable.

**Conflicts of Interest:** The author declares no conflict of interest.

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
