# Peer review of "Changing the Level of Education and Career Choice Depending on the Socioeconomic Status of the Family: Evidence from Azerbaijan"

_sustainability, doi:10.3390/su152215845_

Round 1

Reviewer 1 Report

The article is devoted to the influence of the socio-economic status of families on the level of education in Azerbaijan and is quite relevant in terms of support for increasing attention to the development of human capital as an oil-rich country. However, in my opinion, some corrections are needed in the article.

Ä°n the context of a developing country rich in natural resources, information about the current state and development of education in the country will support the justification of the relevance of the article (I recommend authors to look through this article: Hasanli, Y.; Sadik-Zada, E.R.; Ismayilova, S.; Rahimli, G.; Ismayilova, F. Could the Lacking Absorption Capacity of the Inflowing Capital Be the Real Cause of the Resource Curse?—A Case Study of Transition Economies. Sustainability 202315, 10837. https://doi.org/10.3390/su151410837).

Whether the number of participants in the survey is sufficient to generalize the results should be justified.

Author Response

Thank you for your comments.

Reviewer 2 Report

This work explores the relationship between the socio-economic status of the family and the level of education in Azerbaijan. To this end, the influence of two main independent variables, namely the influence of family elders and family income, on the level of education of an individual was studied. As a result of the study, it was found that mothers, compared to grandparents and grandparents compared to fathers, have a positive impact on the level of education of children. It has been found that household income is a determining factor in choosing a risky career, and children from families with an upper middle-income especially are more positive than those from low-income families.

The contributions of this paper are novel. The title, the keywords and the overall organization of the paper is fine. I like this paper.

(1) The conclusion should include the main findings and implications of the results.

(2) missing related refs that need to back many arguments that need to be supported

(3) The work needs to include some more recent and relevant works in the area of poverty. Some examples of these works are listed below:
--The Nexus between Credit Channels and Farm Household Vulnerability to Poverty: Evidence from Rural China. Sustainability. 2020; 12(7):3019.
-Improving productivity among smallholder farmers in Ghana: does financial inclusion matter?. Agricultural Finance Review, 81(4), 481-502.
--Differences and Influencing Factors of Relative Poverty of Urban and Rural Residents in China Based on the Survey of 31 Provinces and Cities. International Journal of Environmental Research and Public Health. 2022; 19(15):9015.

Author Response

Thank you for your comments. 

Reviewer 3 Report

Dear authors,

Congratulations for the article. It is an interesting and competently analyzed topic. 

Best regards, 

Author Response

Thank you for your comments. 
